# Sterol homeostasis requires regulated degradation of squalene monooxygenase by the ubiquitin ligase Doa10/Teb4

Ombretta Foresti[1,2], Annamaria Ruggiano[1,2], Hans K Hannibal-Bach[3], Christer S Ejsing[3], Pedro Carvalho[1,2]*

[1]Cell and Developmental Biology Programme, Center for Genomic Regulation (CRG), Barcelona, Spain; [2]Universitat Pompeu Fabra, Barcelona, Spain; [3]Department of Biochemistry and Molecular Biology, University of Southern Denmark, Odense, Denmark

**Abstract** Sterol homeostasis is essential for the function of cellular membranes and requires feedback inhibition of HMGR, a rate-limiting enzyme of the mevalonate pathway. As HMGR acts at the beginning of the pathway, its regulation affects the synthesis of sterols and of other essential mevalonate-derived metabolites, such as ubiquinone or dolichol. Here, we describe a novel, evolutionarily conserved feedback system operating at a sterol-specific step of the mevalonate pathway. This involves the sterol-dependent degradation of squalene monooxygenase mediated by the yeast Doa10 or mammalian Teb4, a ubiquitin ligase implicated in a branch of the endoplasmic reticulum (ER)-associated protein degradation (ERAD) pathway. Since the other branch of ERAD is required for HMGR regulation, our results reveal a fundamental role for ERAD in sterol homeostasis, with the two branches of this pathway acting together to control sterol biosynthesis at different levels and thereby allowing independent regulation of multiple products of the mevalonate pathway.

*For correspondence: pedro. carvalho@crg.eu

**Competing interests:** The authors declare that no competing interests exist.

**Reviewing editor**: Michael S Brown, University of Texas Southwestern Medical School, United States

## Introduction

Sterols, such as cholesterol in animals or ergosterol in yeast, are essential components of cellular membranes and their concentration to a large extent determines many of the membrane properties, such as fluidity and rigidity. Therefore, cells evolved sophisticated mechanisms to precisely regulate their sterol levels (*Goldstein et al., 2006*; *Brown and Goldstein, 2009*). These are critical not only for adjusting membrane properties to diverse cellular environments but also for preventing the accumulation of free sterols, which is toxic both to individual cells and to whole organisms (*Goldstein et al., 2006*; *Espenshade and Hughes, 2007*; *Maxfield and van Meer, 2010*).

The regulation of cellular sterol levels occurs primarily during their biosynthesis in the endoplasmic reticulum (ER) by the mevalonate pathway (*Espenshade and Hughes, 2007*; *Brown and Goldstein, 2009*). This highly conserved pathway produces isoprenoids, precursors not only for sterols but also for other essential molecules such as dolichol or ubiquinone (*Figure 1A*; *Goldstein and Brown, 1990*). While a constant supply of these molecules is required, cells must avoid overaccumulation of sterols, a balance that is achieved by a number of feedback systems operating at the transcription, translation, and post-translational levels. Remarkably, decades of work demonstrated that many of these homeostatic control systems converge on the regulation of 3-hydroxy-3-methylglutaryl-coenzyme A reductase (HMGR), an enzyme involved in an early and rate-limiting step of the mevalonate pathway (*Brown and Goldstein, 2009*; *Burg and Espenshade, 2011*).

A key mechanism in sterol homeostasis involves the proteasomal degradation of HMGR in a sterol-dependent manner (*Burg and Espenshade, 2011*). In fact, an increase in sterol biosynthetic intermediates, such as geranylgeranyl pyrophosphate in yeast and lanosterol or its immediate product

**eLife digest** All cells are enclosed by a membrane that is made up of fatty molecules called lipids and is studded with proteins. This membrane allows cells to detect and react to outside events. Since external conditions, such as temperature, can vary dramatically, membranes need to be able to adjust their properties. For example, lipids become more fluid as the temperature rises, so membranes respond to heat stress by incorporating molecules called sterols to increase their rigidity. In fact, sterols have profound effects on membrane properties and are essential to regulate a number of cellular processes. But high levels of sterols can become toxic, so it is essential that they are carefully controlled.

Sterols, such as ergosterol in yeast or cholesterol in mammals, are synthesized in a tightly regulated multi-step process; some of the early steps in sterol production also make common building blocks for other key molecules in the cell. A mechanism to control sterol levels is the regulated destruction of an enzyme that carries out an early step of their synthesis. This occurs via one branch of the ER-associated degradation (ERAD) pathway, which also destroys non-functional proteins. Now, Foresti et al. have found that sterol synthesis is also regulated by another branch of the ERAD pathway. This second control point, which occurs later in the biosynthetic process, allows cells to regulate sterol levels independent of the other products of the pathway that are derived from the same preliminary compounds.

In yeast, the two ERAD branches are directed by Hrd1 and Doa10. These are both ubiquitin ligases—proteins that attach a tag called ubiquitin to other proteins, thus labeling them for recycling by the proteasome (essentially a waste-disposal complex in the cell). To identify the proteins that are tagged by Doa10, Foresti et al. compared protein levels in strains lacking Doa10 with those in wild type yeast. Unexpectedly, the enzyme Erg1, which helps to synthesize ergosterol, was more abundant in cells lacking Doa10.

Foresti et al. found that Doa10 tagged Erg1 for destruction when levels of the building blocks of ergosterol rose inside the cell. These ergosterol intermediates are toxic to yeast, which converts them into less harmful molecules known as sterol esters using the proteins Are1/2. When the *DOA10* or *ARE1/2* genes were deleted, these intermediates were more abundant; strikingly, they became even more prevalent when all three genes were knocked out in the same strain. In contrast, blocking the other ERAD branch by deleting *HRD1* did not cause ergosterol intermediates to accumulate, nor did it exacerbate the effects of *ARE1/2* knockout.

When combined with previous findings, these results provide evidence that the different branches of the ERAD pathway regulate ergosterol synthesis at distinct steps. The same mechanism is observed in human cells when high levels of cholesterol are detected. By identifying parallel routes to control sterol levels, this work reinforces the importance of membrane integrity to life.

24,25-dihydrolanosterol in mammals, strongly accelerates the degradation of HMGR, thereby lowering the flux through the mevalonate pathway. Both in yeast and in mammals, the targeting of HMGR to the proteasome for degradation is mediated by a branch of the ER-associated protein degradation (or ERAD) pathway, which is primarily studied for its role in the elimination of misfolded ER proteins (**Smith et al., 2011**; **Brodsky, 2012**). Importantly, ERAD factors involved in the recognition of misfolded ER proteins are not required for HMGR degradation. Instead specific chaperones called Insigs, in the presence of the right sterol signal, regulate the interaction of HMGR with the central ERAD components, a ubiquitin ligase complex in the ER membrane, called Hrd1 in yeast and Gp78 in mammals (**Hampton et al., 1996**; **Bays et al., 2001**; **Sever et al., 2003**; **Flury et al., 2005**; **Song et al., 2005b**). The Hrd1/Gp78-dependent ubiquitination of HMGR leads to its membrane extraction, facilitated by the Cdc48/p97 ATPase, and release in the cytoplasm for degradation by the proteasome. In mammalian cells, a second ER-bound ubiquitin ligase, Trc8, can also promote the sterol-dependent degradation of HMGR (**Jo et al., 2011a**). Other branches of ERAD have never been implicated in sterol homeostasis.

Here, we report a novel role of ERAD in the regulation of sterol biosynthesis. A screen for substrates of the ERAD ubiquitin ligase Doa10 identified the squalene monooxygenase Erg1, an enzyme required for a sterol-specific step of the mevalonate pathway in *Saccharomyces cerevisiae*. We show that Doa10-dependent degradation of Erg1 is regulated by the levels of lanosterol and that, together

with sterol esterification, it is essential for preventing the accumulation of toxic sterol intermediates. Moreover, in mammalian cells, sterol-dependent degradation of squalene monooxygenase requires the Doa10 homologue Teb4. Altogether, our findings reveal an evolutionarily conserved, central role of ERAD in sterol homeostasis.

## Results

### Erg1 is a substrate of the Doa10 complex

To identify novel substrates of the ERAD ubiquitin ligase Doa10 (*Swanson et al., 2001*), we used SILAC (Stable Isotope Labeling by Amino acids in Culture) labeling followed by quantitative proteomics (*de Godoy et al., 2008*). We found that, at steady state, several proteins were overrepresented in *doa10Δ* mutant when compared to wild type (*wt*) cells (data not shown). Among these potential Doa10 substrates was the yeast squalene monooxygenase Erg1 (*Figure 1A, B*). Other components of the yeast sterol biosynthetic pathway (ergosterol or Erg pathway) were present at similar levels in *doa10Δ* and *wt* cells (*Figure 1—figure supplement 1A*). The high levels of Erg1 in *doa10Δ* mutants are not due to increased *ERG1* transcription as the abundance of Erg1 mRNA was indistinguishable between *doa10Δ* and *wt* cells (data not shown). Interestingly, cells lacking Ubc6 or Ubc7, the ubiquitin-conjugating enzymes required for Doa10-dependent ubiquitination and members of the Doa10 complex (*Carvalho et al., 2006*), also showed increased steady state levels of Erg1 when compared to *wt* cells (*Figure 1B*). In contrast, deletion of Hrd1, Der1, or Usa1, involved in a different branch of ERAD as part of the Hrd1 complex (*Carvalho et al., 2006*), had no effect on the steady state levels of Erg1 (*Figure 1B*). To directly test for a role of Doa10 in the turnover of Erg1, we performed cycloheximide shut-off experiments. We found that in *wt* cells Erg1 is an unstable protein with a half-life of <120 min (*Figure 1C*). Deletion of the ubiquitin ligase Doa10 or any of the components of the Doa10 complex strongly impaired the degradation of Erg1 expressed from its own promoter (*Figure 1C*) or from the heterologous glyceraldehyde-3-phosphate dehydrogenase (*GAPDH*) promoter (*Figure 1—figure supplement 1B*). Similarly, Erg1 degradation was virtually blocked in *cdc48-3* and *npl4-1* cells, expressing temperature sensitive alleles in essential subunits of the Cdc48 ATPase complex that pulls substrates out of the ER membrane after Doa10-dependent ubiquitination (*Figure 1D*). On the other hand, in *hrd1Δ* cells the kinetics of Erg1 degradation was similar to that in *wt* cells (*Figure 1C* and *Figure 1—figure supplement 1B*). Together, these data show that Erg1 is an ERAD substrate of the Doa10 complex.

### ERAD of Erg1 depends on a single lysine residue

In the vast majority of cases, protein ubiquitination occurs on the ε-amino group of lysine residues. We therefore searched for Erg1 lysine residues required for its Doa10-dependent degradation. We focused our attention on a cluster of residues proximal to the C-terminal region (K278, K284, K311, and K360) that were reported to be ubiquitinated in several large scale studies (*Hitchcock et al., 2003*; *Peng et al., 2003*; *Beltrao et al., 2012*). We generated Erg1-derivatives in which individual or pairs of the lysine residues were mutated to arginine. These lysine mutant alleles supported the growth of yeast cells, indicating that the substitutions did not significantly affect the essential enzymatic function of Erg1 (*Figure 2A*). To test the effect of the lysine mutations on Erg1 stability, we performed cycloheximide shut-off assays in otherwise *wt* cells. Degradation of Erg1(K278,284R) and Erg1(K360R) was indistinguishable from degradation of *wt* Erg1 (*Figure 2B, C*). In contrast, Erg1(K311R) was strongly stabilized either when expressed from the endogenous *ERG1* promoter (*Figure 2B, C*) or from the strong constitutive *GAPDH* promoter (*Figure 2—figure supplement 1A*) . Importantly, Erg1(K311R) and Erg1(K278,284,311,360R), with simultaneous mutations on four lysine residues, were only slightly stabilized by additional deletion of *DOA10* (*Figure 2B, C*). Thus, the lysine residue at position 311 is essential for Doa10-dependent degradation of Erg1.

### Lanosterol levels regulate Erg1 degradation

We then tested whether Doa10-dependent degradation of Erg1 was affected by the levels of cellular sterols. To manipulate the levels of sterols and sterol biosynthetic intermediates, we took advantage of several well characterized small molecule inhibitors of the Erg pathway enzymes (*Figure 3A*). Reduction of ergosterol synthesis in *wt* cells by a brief treatment with zaragozic acid, an inhibitor of the squalene synthetase Erg9, led to a strong stabilization of Erg1 when compared to controls (*Figure 3B*). Similar treatments did not affect the degradation kinetics of the Doa10 substrates Vma12-Ndc10$_{902–956}$-HA and Pca1$_{1–392}$-DHFR-HA (*Figure 3C* and *Figure 3—figure supplement 1A–C*) (*Adle et al., 2009*;

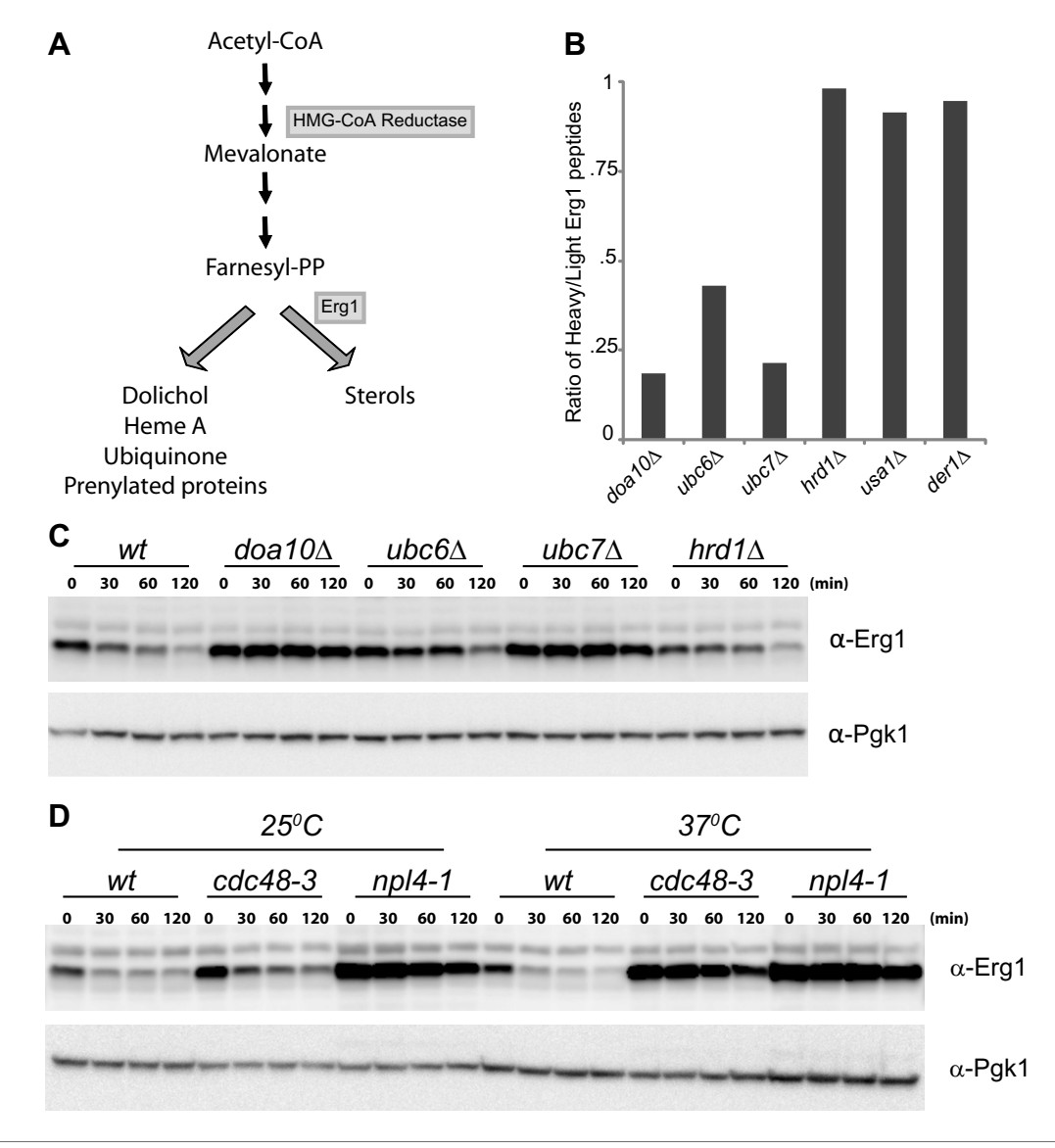

**Figure 1**. Erg1 is a substrate of the Doa10 complex. (**A**) Schematic representation of the mevalonate pathway and its different end products. The steps catalyzed by HMG-CoA reductase and the squalene monooxygenase Erg1 are indicated. Adapted from ***Goldstein and Brown (1990)***. (**B**) Erg1 abundance in the indicated mutants relative to *wt* cells, as detected by mass spectrometry upon SILAC labeling. All strains used are lysine auxotrophs and were grown in the presence of either heavy L-lysine (*wt* cells) or light L-lysine (deletion mutants). Note that high steady state levels of Erg1 result in a low heavy/light ratio. (**C**) The degradation of endogenous Erg1 was followed after inhibition of protein synthesis by cycloheximide in *wt* cells or in cells with the indicated deletions. Whole-cell extracts were analyzed by SDS–PAGE and western blotting. Erg1 was detected with α-Erg1 antibody. Phosphoglycerate kinase (Pgk1) was used as loading control and detected with α-Pgk1 antibodies. A representative gel of three independent experiments is shown. (**D**) The degradation of endogenous Erg1 was analyzed as in (**C**) in *wt* cells or the temperature sensitive *cdc48-3* and *npl4-1* cells either at the permissive temperature of 25°C or after a 2 hr shift to 37°C, the restrictive temperature.

The following figure supplements are available for figure 1:

**Figure supplement 1**. Abundance of Erg1 but not of other components of the Erg pathway is altered in Doa10 complex mutants.

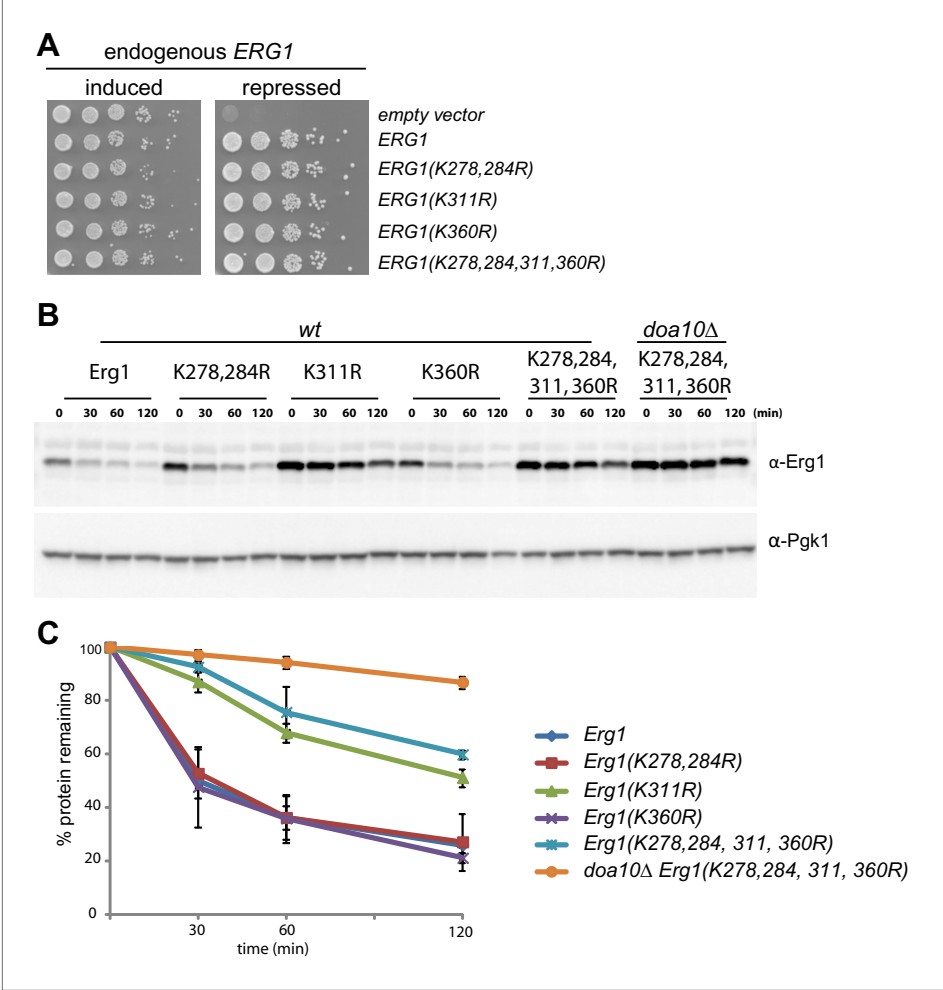

**Figure 2**. Doa10-dependent degradation of Erg1 depends on a single lysine residue. (**A**) Expression of *ERG1* or *ERG1*-derivatives with the indicated lysine mutations from a plasmid rescues the growth of yeast cells upon repression of the endogenous *ERG1*. A yeast strain expressing endogenous *ERG1* from the regulated *GAL1* promoter was transformed with plasmids encoding different lysine mutants or a control empty vector. The growth of serial dilutions of cells was tested under conditions of induced (galactose-containing media) or repressed (glucose-containing media) endogenous *ERG1*. (**B**) The degradation of Erg1 or the indicated Erg1 lysine mutant expressed from the endogenous *ERG1* promoter was followed after inhibition of protein synthesis by cycloheximide in *wt* or *doa10Δ* cells. Samples were analyzed as in *Figure 1C*. (**C**) Quantitation of two independent experiments performed as described in (**B**).

The following figure supplements are available for figure 2:

**Figure supplement 1**. Doa10-dependent degradation of Erg1 depends on the lysine residue at position 311.

*Furth et al., 2011*). These experiments indicate that sterol depletion specifically affects the Doa10-dependent degradation of Erg1 but not other Doa10 substrates, such as misfolded proteins. Similar results were obtained upon treatment of *wt* cells with Ro48-807, an inhibitor of the squalene cyclase Erg7 and whose product (lanosterol) is the first sterol of the pathway (*Figure 3A,D,E* and *Figure 3—figure supplement 1D–F*). The next step in ergosterol biosynthesis is catalyzed by Erg11, the lanosterol demethylase, which is inhibited by fluconazole (*Figure 3A*). In contrast to zaragozic acid and Ro48-807, a short treatment of *wt* cells with fluconazole induced a marked acceleration of Erg1 degradation (*Figure 3F,G*). Importantly, the acceleration of Erg1 degradation upon fluconazole treatment was completely dependent on Doa10, as it was blocked in *doa10Δ* cells (*Figure 3F*). Moreover, fluconazole treatment did not have any effect on the kinetics of degradation of the Doa10 substrates Pca1$_{1–392}$-DHFR-HA (*Figure 3G* and *Figure 3—figure supplement 1G*) and

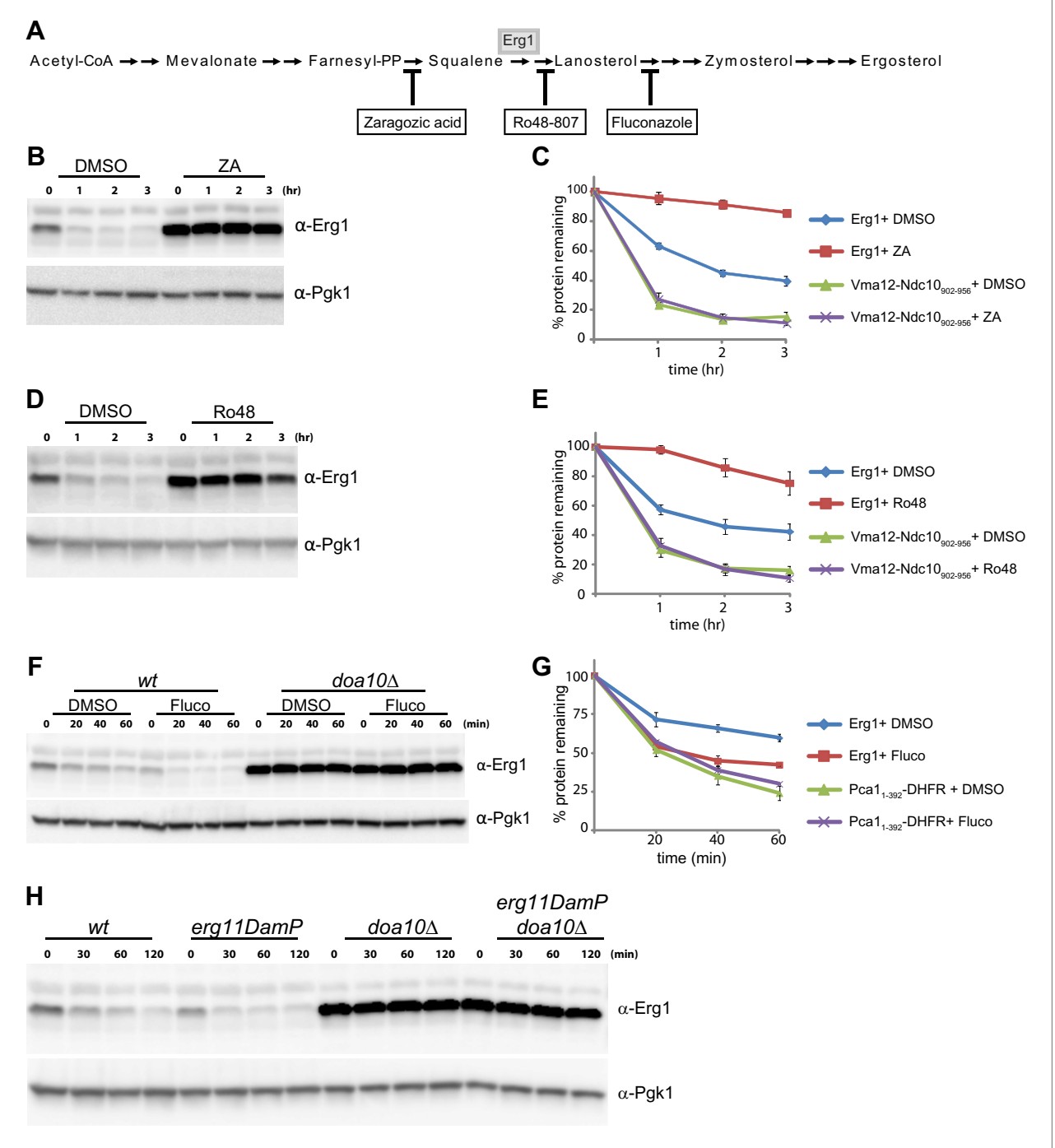

**Figure 3**. Flux through the sterol pathway regulates Erg1 degradation. (**A**) Schematic representation of the ergosterol biosynthetic pathway highlighting the enzymatic steps affected by the small inhibitor zaragozic acid, Ro48-807, and fluconazole. The step catalyzed by the squalene monooxygenase Erg1 is also indicated. (**B**) The degradation of endogenous Erg1 was followed after inhibition of protein synthesis by cycloheximide in *wt* control cells (DMSO) or *wt* cells treated for 2 hr with 10 μg/ml zaragozic acid (ZA). Samples were analyzed as described in *Figure 1C*. (**C**) Quantitation of three independent experiments as described in (**B**) in cells expressing the Doa10 substrate Vma12-Ndc10$_{902–956}$-HA, a model misfolded protein. Vma12-Ndc10$_{902–956}$-HA was detected with anti-HA antibodies. (**D**) The degradation of endogenous Erg1 was followed after inhibition of protein synthesis by cycloheximide in *wt* control cells (DMSO) or *wt* cells treated for 2 hr with 40 μg/ml Ro48-807 (Ro48). Samples were analyzed as described in *Figure 1C*. (**E**) Quantitation of three independent experiments as described in (**D**) in cells expressing the Doa10 substrate Vma12-Ndc10$_{902–956}$-HA, a model misfolded protein. Vma12-Ndc10$_{902–956}$-HA was detected with anti-HA antibodies. (**F**) The degradation of endogenous Erg1 was followed after inhibition of protein synthesis by cycloheximide in *wt* and *doa10Δ* cells. Cells were incubated for 1 hr with DMSO or with 10 μg/ml fluconazole (Fluco). Samples were analyzed as described in *Figure 3. Continued on next page*

*Figure 3. Continued*

**Figure 1C**. (**G**) Quantitation of three independent experiments as described in (**F**) in cells expressing the Doa10 substrate Pca1$_{1–392}$-DHFR-HA, a model misfolded protein. Pca1$_{1–392}$-DHFR-HA was detected with anti-HA antibodies. (**H**) The degradation of endogenous Erg1 was followed after inhibition of protein synthesis by cycloheximide in *wt* cells or in cells with the indicated mutations. Samples were analyzed as described in **Figure 1C**.
The following figure supplements are available for figure 3:

**Figure supplement 1**. Sterol depletion affects Doa10-dependent degradation of Erg1 but not of other Doa10 substrates.

Vma12-Ndc10$_{902–956}$-HA (**Figure 3—figure supplement 1H**). Therefore, lanosterol accumulation induced by fluconazole treatment stimulates Doa10-dependent degradation of Erg1. In agreement with these results, we found that Erg1 degradation was also accelerated in a *erg11DAmP* mutant bearing a hypomorphic allele of *ERG11* (**Figure 3H**). Mutation of genes required for the very last steps of ergosterol biosynthesis (*ERG2* to *ERG6*) did not have a major impact on the kinetics of Erg1 degradation (**Figure 3—figure supplement 1I**). Taken together, these data demonstrate that Doa10-dependent degradation of Erg1 is regulated by the levels of sterols, most likely lanosterol.

## *doa10Δ* cells accumulate sterol precursors

We then asked whether the Doa10-dependent degradation of Erg1 had an effect on sterol homeostasis. To address this issue, we analyzed the lipid composition of *doa10Δ* cells by shotgun lipidomics (**Ejsing et al., 2009**). While the overall sterol levels in *doa10Δ* and *wt* cells were comparable, the relative abundance of individual sterol species was significantly different (**Figure 4A**). When compared to *wt* cells, the *doa10Δ* mutant showed a reduction in the levels of ergosterol (by 13%) with a concomitant fivefold increase in the ergosterol precursor lanosterol. Moreover, *doa10Δ* cells had small amounts of ergostadienol, a sterol intermediate virtually undetectable in *wt* cells (**Figure 4A**). Ergosterol concentration is a major determinant of membrane fluidity, which is tightly regulated according to the cellular environment (**Brown and Goldstein, 2009**; **Burg and Espenshade, 2011**). The changes in sterol composition observed in *doa10Δ* mutants are expected to affect the membrane properties in these cells. Accordingly, we noticed that mutations in components of the Doa10 complex have a mild growth phenotype when grown at low temperatures (10°C), as previously reported (**Loertscher et al., 2006** and data not shown). Altogether, the data presented so far indicate that sterol-dependent degradation of Erg1 by Doa10 is part of a feedback system essential for sterol homeostasis.

## Parallel pathways monitoring buildup of sterol intermediates

A mechanism preventing the accumulation of sterol intermediate metabolites, which are toxic, is their esterification into sterol esters (**Chang et al., 2006**; **Jacquier and Schneiter, 2012**). Interestingly, *doa10Δ* cells also accumulated significantly higher amounts of sterol esters (approximately a 40% increase in relation to *wt* cells; **Figure 4B**). This increase prompted us to investigate a potential connection between the Doa10-dependent feedback regulation of Erg1 and sterol esterification. In yeast, esterified sterols are produced by two partially redundant acyl-CoA sterol acyl transferases (ASATs), Are1 and Are2, that are not essential for growth under laboratory conditions (**Yang et al., 1996**; **Zweytick et al., 2000**). We then again used shotgun lipidomics to evaluate the effect of the *DOA10* mutation on sterol composition in esterification-deficient *are1Δ are2Δ* cells. Compared to *wt* cells, *are1Δ are2Δ* mutant has ~20% less ergosterol (**Figure 4A**), as previously reported (**Yang et al., 1996**; **Zweytick et al., 2000**). Moreover, *are1Δ are2Δ* cells show significantly higher amounts of lanosterol and ergostadienol (**Figure 4A**), indicating that loss of esterification also leads to the accumulation of sterol intermediates. Strikingly, in *are1Δ are2Δ doa10Δ* mutant the levels of ergosterol dropped by more than 50%, while the levels of the intermediates lanosterol and ergostadienol increased dramatically (**Figure 4A**). The changes observed in the *are1Δ are2Δ doa10Δ* mutant, although of a much larger magnitude, perfectly mirror those observed in *doa10Δ* cells. Importantly, the effects appear specific to *doa10Δ*, as deletion of *HRD1* did not significantly change the sterol profile of esterification-deficient cells (data not shown). These data suggest that sterol esterification and the ubiquitin ligase Doa10 play parallel, redundant functions in preventing accumulation of sterol intermediate metabolites. Does the massive increase in sterol intermediates at the expense of the final product, ergosterol, have any impact in the cellular physiology of *are1Δ are2Δ doa10Δ* cells? To address this issue, we have performed growth assays under conditions of membrane stress such as low temperature or the presence of benzyl

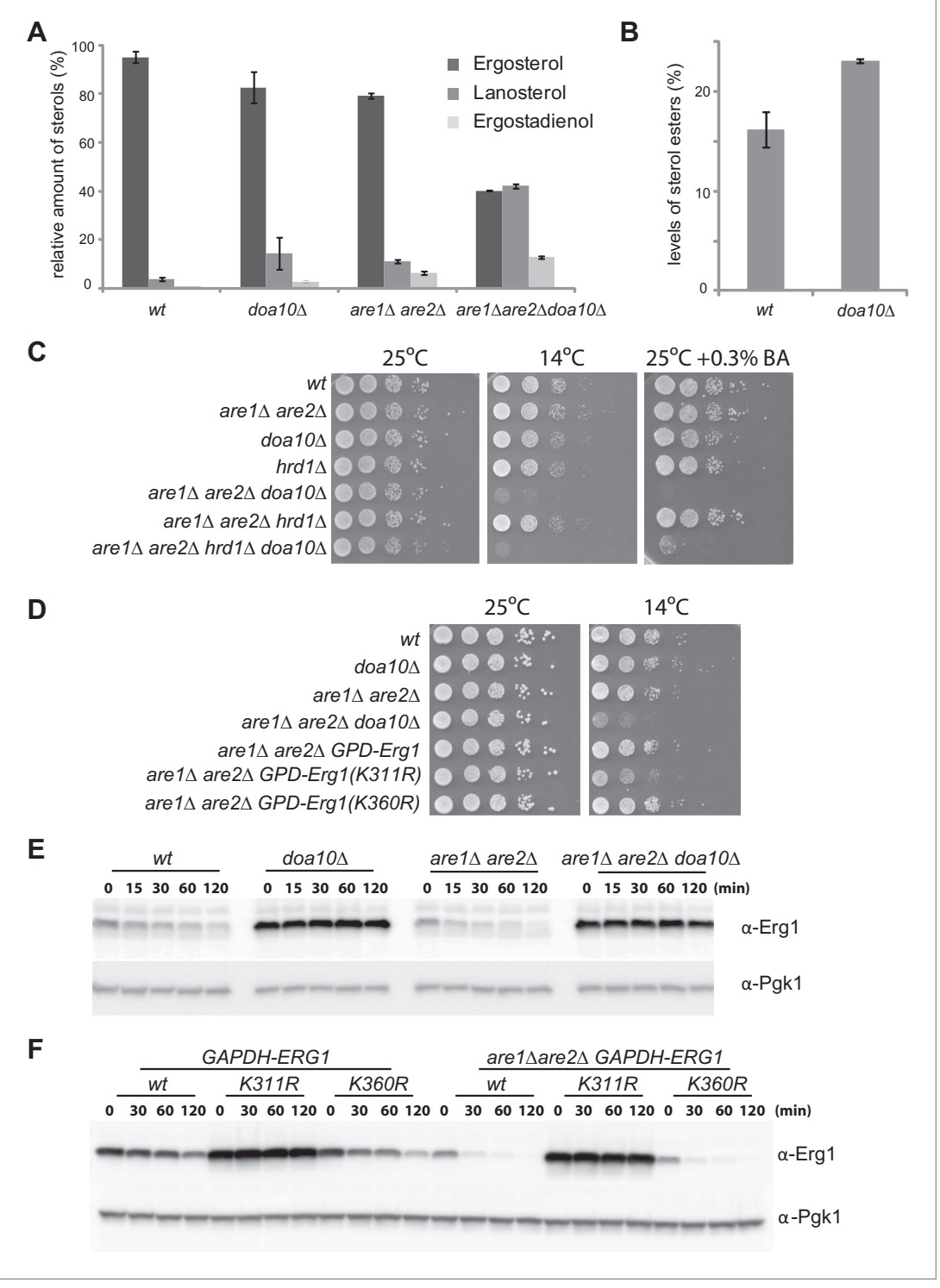

**Figure 4**. Doa10-dependent degradation of Erg1 affects sterol homeostasis and, together with sterol esterification, is essential to prevent buildup of sterol intermediates. (**A**) Relative amounts of ergosterol, lanosterol, and ergostadienol in cells with the indicated genotype. Cells were grown in synthetic complete (SC) media until early stationary phase, and lipids were extracted and analyzed by shotgun lipidomics. (**B**) Analysis of sterol esters in *wt* and *doa10Δ* cells by mass spectrometry. Cells were grown in SC media until early stationary phase, and lipids were extracted and analyzed by shotgun lipidomics. (**C**) Serial dilutions of cells with the indicated genotype were spotted on YPD or YPD + 0.3% of benzyl alcohol (BA) and incubated for 2 (25°C), 4 (YPD + 0.3% BA) or 5 days (14°C). (**D**) Serial dilutions of cells with

*Figure 4. Continued on next page*

*Figure 4. Continued*

the indicated genotype were spotted on YPD and incubated for 2 (25°C) or 5 days (14°C). (**E**) The degradation of endogenous Erg1 was followed after inhibition of protein synthesis by cycloheximide in cells with the indicated genotype. Samples were analyzed as described in **Figure 1C**. (**F**) The degradation of Erg1 or the indicated Erg1 lysine mutants expressed from the constitutive glyceraldehyde-3-phosphate dehydrogenase (*GAPDH*) promoter was followed after inhibition of protein synthesis by cycloheximide in *wt* or *are1Δ are2Δ* cells. Samples were analyzed as described in **Figure 1C**.

alcohol, a membrane fluidizing agent. Although *are1Δ are2Δ doa10Δ* mutant grows normally at 25°C, these cells barely grew when incubated at 14°C or in the presence of benzyl alcohol (**Figure 4C**). The same phenotype was observed when *are1Δ are2Δ* mutation was combined with deletion of *UBC7*, encoding for the ubiquitin-conjugating enzyme in complex with Doa10 (data not shown). In contrast, the *are1Δ are2Δ hrd1Δ* mutant cells, lacking the ubiquitin ligase of the other ERAD branch, did not display any growth defect at 14°C or in the presence of benzyl alcohol (**Figure 4C**). These results indicate that accumulation of sterol intermediates with a concomitant decrease in ergosterol levels renders *are1Δ are2Δ doa10Δ* cells vulnerable to membrane stress, perhaps due to the inability to adjust membrane fluidity. To rule out the possibility that the growth phenotypes of *are1Δ are2Δ doa10Δ* are due to pleiotropic effects of *doa10Δ*, for example due to its role in the clearance of misfolded proteins, we took advantage of the degradation resistant *ERG1* allele, *ERG1(K311R)*. Remarkably, constitutive expression from the *GAPDH* promoter of *ERG1(K311R)* in *are1Δ are2Δ* cells led to a cold phenotype comparable to that observed in *are1Δ are2Δ doa10Δ* mutant (**Figure 4D**). In contrast, expression of *ERG1* or *ERG1(K360R)* under the same conditions did not cause any apparent defect (**Figure 4D**), suggesting that Erg1 is a key Doa10 substrate for sterol homeostasis. In sum, these data show that Are1, Are2-dependent sterol esterification and Doa10-dependent degradation of Erg1 are parallel mechanisms preventing abnormal accumulation of potentially toxic sterol intermediates. Interestingly, these two mechanisms appear to be tightly integrated in cells. As suggested by previous studies (**Sorger et al., 2004**), we found that degradation of Erg1 is accelerated in *are1Δ are2Δ* cells (**Figure 4E, F**), likely due to the higher levels of lanosterol in this mutant (**Figure 4A**). Importantly, this accelerated degradation still requires the lysine at position 311 (**Figure 4F**) and is completely dependent on Doa10 as it is blocked in *doa10Δ* cells (**Figure 4E**).

## Doa10 homologue Teb4 promotes degradation of human squalene monooxygenase SM

The homologue of Erg1 in mammals, SM, is ubiquitinated and degraded by the proteasome (**Gill et al., 2011**). Interestingly, SM proteasomal-dependent degradation is stimulated by sterols, in this case by the final product of the pathway, cholesterol (**Gill et al., 2011**). These similarities prompted us to test whether the mammalian homologue of Doa10, the ubiquitin ligase Teb4 (**Swanson et al., 2001**; **Hassink et al., 2005**), was involved in the turnover of SM. We transfected human embryonic kidney (HEK) 293 cells with a pool of siRNA against Teb4, which led to a 57% (±0.044%) reduction in TEB4 mRNA levels compared to control treated cells, as detected by qPCR. We noticed that although transcription of SM in Teb4-depleted and control cells was indistinguishable, the steady state levels of SM were 1.8-fold (±0.232) higher in cells treated with Teb4 siRNA (**Figure 5A**, 'untreated' lanes). This stabilization is consistent with a role of Teb4 in the degradation of SM. To directly examine SM half-life upon Teb4 depletion we performed cycloheximide shut-off experiments with or without the addition of cholesterol to stimulate SM degradation. These experiments were performed in sterol-deprived cells, as previously described (**Gill et al., 2011**; **Jo et al., 2011b**). In these conditions, SM appears to have a relatively long half-life that is even longer in Teb4 siRNA-treated cells (**Figure 5A, B**). Cholesterol treatment of control cells induces very rapid degradation of SM (half-life <4 hr), as previously shown (**Gill et al., 2011**). In contrast, cholesterol treatment in Teb4-depleted cells has a much milder effect on the degradation of SM (**Figure 5A,B**) and its half-life remains longer than 4 hr (**Figure 5B**). Both in control and in Teb4-depleted cells, SM sterol-dependent degradation is significantly attenuated by the proteasome inhibitor MG132 (**Figure 5A**). To independently assess the role of Teb4 in the regulated degradation of SM, HEK293 cells were transfected with constructs overexpressing Teb4-myc or Teb4(C9A)-myc, a dominant negative mutant that lacks ubiquitin ligase activity (**Hassink et al., 2005**). While SM degradation occurred with normal kinetics in Teb4-overexpressing cells, expression of Teb4(C9A) strongly inhibited the cholesterol-dependent acceleration of SM degradation, as assayed

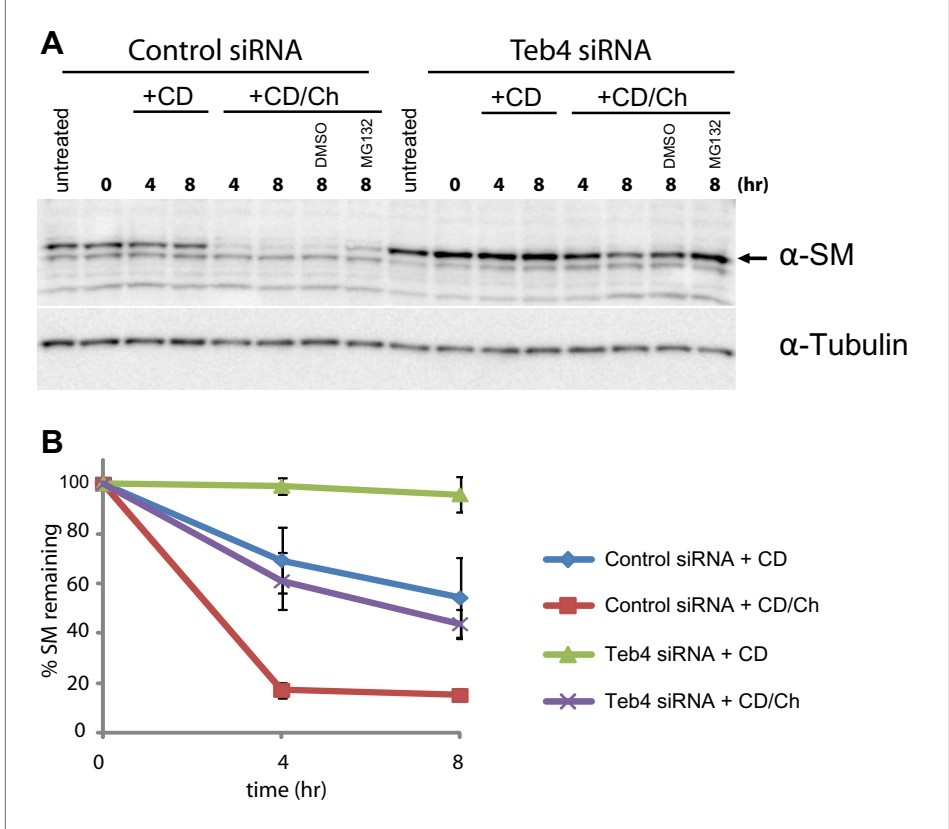

**Figure 5**. Doa10 homologue Teb4 promotes degradation of human squalene monooxygenase. (**A**) The degradation of endogenous squalene monooxygenase (SM) was followed after inhibition of protein synthesis by cycloheximide in sterol-deprived HEK293 cells treated with control siRNA or siRNA targeting the ubiquitin ligase Teb4. The degradation of SM was monitored under basal conditions (methyl-β-cyclodextrin vehicle, +CD) and upon addition of cholesterol (+CD/Ch). Where indicated DMSO or the proteasome inhibitor MG132 (10 µM) was included. Samples were analyzed by SDS–PAGE and immunoblotting. SM was detected with rabbit polyclonal anti-SM antibodies and α-tubulin with mouse monoclonal anti-α-tubulin antibodies. The arrow indicates the band corresponding to SM. (**B**) Quantitation of three independent experiments performed as described in (**A**).

The following figure supplements are available for figure 5:

**Figure supplement 1**. Overexpression of dominant negative Teb4(C9A), but not of *wt* Teb4, strongly delays degradation of human squalene monooxygenase.

by cycloheximide shut-off experiments (*Figure 5—figure supplement 1*). Altogether, these data indicate that the Teb4 ubiquitin ligase promotes the degradation of SM in a cholesterol-dependent manner. Moreover, it emphasizes the remarkable conservation of the role of ERAD in regulating sterol homeostasis in yeast and mammals.

## Discussion

Here we characterize a novel role of the ERAD pathway in the homeostatic control of sterol levels and demonstrate that loss of this feedback system leads to sterol deregulation. We show that the yeast ubiquitin ligase Doa10, along with the other components of the Doa10 complex, promotes the regulated degradation of the squalene monooxygenase Erg1. The Doa10-mediated degradation of Erg1 is stimulated by lanosterol and loss of Doa10 leads to accumulation of lanosterol and sterol esters. Moreover, we found that the requirement for Doa10 function increases in sterol esterification-deficient cells, indicating that feedback regulation of Erg1 and sterol esterification are redundant pathways preventing accumulation of sterol intermediates.

Lanosterol is the first sterol produced by the pathway and, in mammalian cells, its levels or the levels of an immediate product (24,25-dihydrolanosterol) are key in the regulation of sterol biosynthesis. Specifically, an increase in lanosterol/24,25-dihydrolanosterol levels stimulates the Gp78-dependent ubiquitination of HMGR, its proteasomal degradation, and consequently a decrease in the rate of sterol biosynthesis (*Song et al., 2005a*; *Lange et al., 2008*; *Nguyen et al., 2009*). We have now found that in yeast, high levels of lanosterol also lead to a decrease in flux through the sterol pathway, in this case by the Doa10-dependent degradation of Erg1. Together these observations suggest that the regulation of the rate of sterol biosynthesis according to the lanosterol levels might be a unifying principle of sterol homeostasis. Besides being the first sterol of the pathway, lanosterol is also a poor substrate for the sterol esterification enzyme Are2, the most active of the yeast ASATs (*Zweytick et al., 2000*). Therefore, lanosterol levels might provide an accurate measurement of the flux into the sterol branch of the mevalonate pathway (*Zweytick et al., 2000*; *Espenshade and Hughes, 2007*).

High Erg1 levels in *doa10Δ* cells lead to accumulation of lanosterol, that is particularly prominent in cells also lacking the ASATs Are1 and Are2. This huge increase in lanosterol levels is somewhat surprising because these cells have an intact Erg11, the lanosterol 14-α-demethylase that consumes lanosterol. This indicates that Erg11 activity is either rate limiting or regulated. Erg11 is an enzyme of the cytochrome P450 family that uses heme as a co-factor. Therefore, it is possible that lanosterol accumulation in these mutants is a consequence of changes in the availability of heme, also a product of the mevalonate pathway. Alternatively, the lanosterol accumulation might result from a still unknown regulatory system acting on Erg11 itself or another downstream enzyme. This possibility is supported by the observation that simultaneous overexpression of Erg1 and Erg11 leads to a significant increase in the production of ergosterol in yeast (*Veen et al., 2003*).

Besides the Doa10-dependent degradation of Erg1 in yeast, we show that the Doa10 homologue Teb4 also promotes the degradation of SM in mammalian cells. While the degradation of yeast Erg1 is accelerated by the sterol intermediate lanosterol, SM degradation is stimulated by the end product of the sterol pathway, cholesterol (*Gill et al., 2011*). Moreover, Doa10-dependent degradation of Erg1 requires a single lysine residue that is poorly conserved outside fungi, while proteasomal-dependent degradation of SM depends on an N-terminal fragment conserved only in higher animals (*Gill et al., 2011*). Despite these differences, it appears that the post-translation feedback regulation of squalene monooxygenase by the Doa10/Teb4 ubiquitin ligase is a conserved mechanism of sterol homeostasis. In fact, this regulatory system might also operate in plants, as suggested by the observation that mutations of *Arabidopsis thaliana* Doa10 homologue suppress a hypomorphic allele of squalene monooxygenase (*Doblas et al., 2013*).

HMGR, the major target for Hrd1/Gp78 regulation, acts at a very early step of the mevalonate pathway and is required for the synthesis of all its products (*Figure 1A*). On the other hand, squalene monooxygenase is required for the synthesis of sterols but not of other key metabolites of the mevalonate pathway, such as dolichol or ubiquinone (*Figure 1A*). Therefore, the novel feedback system described here allows cells to independently regulate the biosynthesis of the different metabolites derived from mevalonate. Remarkably, regulation of HMGR and squalene monooxygenase is mediated by two branches of the ERAD pathway, both in yeast and in mammals. As previously shown, the ubiquitin ligase Hrd1/Gp78 promotes the degradation of HMGR (*Bays et al., 2001*; *Song et al., 2005b*). Our results now show that the ubiquitin ligase Doa10/Teb4 promotes the degradation of squalene monooxygenase, which place ERAD at the center of cellular sterol homeostasis, with multiple branches of ERAD acting together to regulate sterol biosynthesis at different levels (*Figure 6*). More broadly, our findings raise the possibility that regulated degradation of folded, active proteins might be a more prominent feature of ERAD, a pathway primarily studied for its role in the elimination of misfolded proteins. Whether regulated ERAD substrates are all related to sterol homeostasis or ERAD plays a more general role in regulating the ER proteome remains to be determined.

Gp78-mediated degradation of HMGR involves proteins called Insigs (*Song et al., 2005b*; *Espenshade and Hughes, 2007*). These proteins function as adaptors: in a sterol-dependent manner they bind to the sterol-sensing domain of HMGR and recruit Gp78 ligase complex (*Song et al., 2005b*). However Insigs are not required for the degradation of squalene monooxygenase either in yeast (AR and PC unpublished results) or in mammals (*Gill et al., 2011*). Therefore, future studies should elucidate the mechanisms by which squalene monooxygenase is recognized by the ubiquitin ligase Doa10/Teb4 in a sterol-dependent manner.

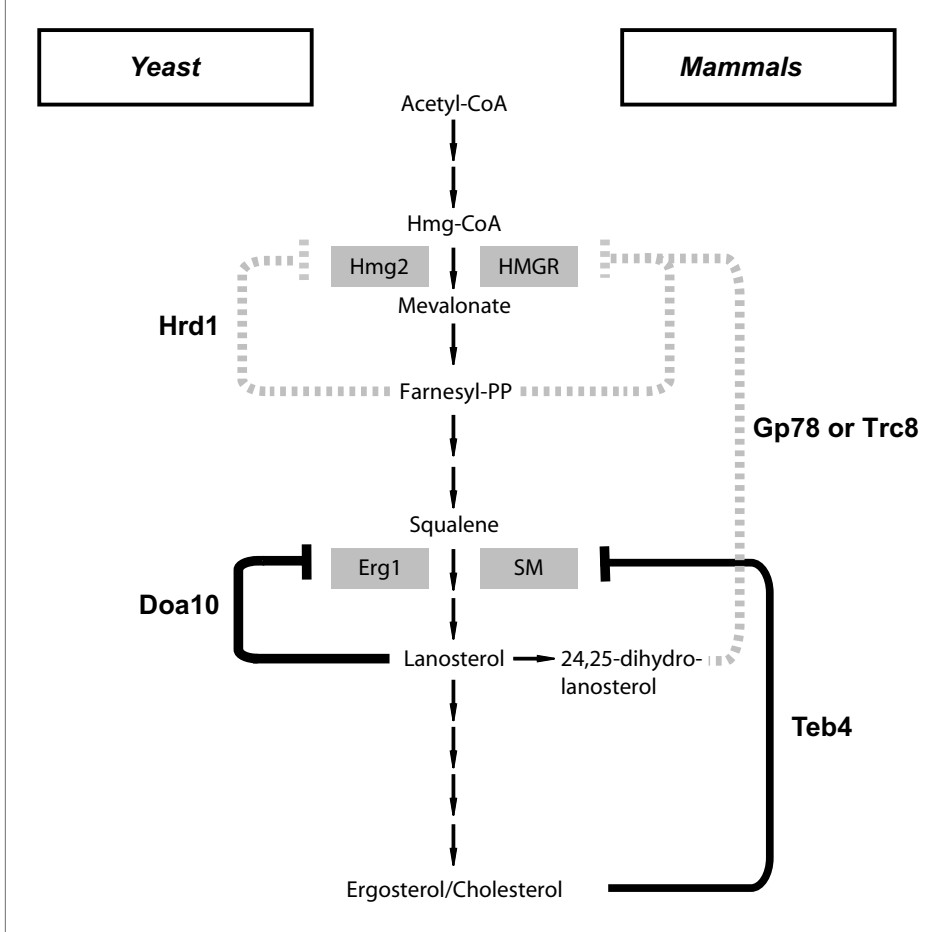

**Figure 6**. A central role of endoplasmic reticulum-associated protein degradation in sterol homeostasis. (**A**) Schematic representation of the feedback inhibition systems required for sterol homeostasis in yeast (left) and mammals (right) previously characterized (dotted lines) and described here (solid lines). Endoplasmic reticulum-associated protein degradation (ERAD) ubiquitin ligases are in bold and the enzymes targeted by ERAD-regulated degradation are enclosed in gray boxes.

## Materials and methods

### Reagents

Heavy lysine [$^{13}C_6$$^{15}N_2$] was purchased from Sigma-Aldrich (St Louis, MO). Cycloheximide (CHX; Sigma-Aldrich) stock was stored at 12.5 mg/ml in water at −20°C and was used at 250 µg/ml. Zaragozic acid (Sigma-Aldrich) stock was stored at 10 mg/ml in DMSO at 20°C and used at 10 µg/ml. Ro48-8071 (Sigma-Aldrich) stock was stored at 40 mg/ml in DMSO and used at 40 µg/ml. Fluconzole (Sigma-Aldrich) stock was stored at 5 mg/ml in DMSO and used at 10 µg/ml. Rat monoclonal anti-HA high-affinity antibody (3F10) was purchased from Roche. Mouse monoclonal anti-phosphoglycerate kinase (Pgk1) was purchased from Invitrogen. The polyclonal rabbit anti-Erg1 antibody was a generous gift from Dr Chao-Wen Wang, Academia Sinica, Taipei, Taiwan. Mouse monoclonal antibodies against human α-tubulin were purchased from Sigma-Aldrich. Rabbit polyclonal antibodies against human SM were purchased from Proteintech (Chicago, IL). All other reagents and chemicals were purchased from Sigma-Aldrich unless otherwise stated.

### Yeast strains and plasmids

Tagging of proteins and individual gene deletions were performed by standard PCR-based homologous recombination (*Longtine et al., 1998*). Strains with multiple gene deletions and/or genomically encoded fusion proteins were made by PCR-based homologous recombination (*Longtine et al., 1998*) or by

crossing haploid cells of opposite mating types, followed by sporulation and tetrad dissection using standard protocols (*Guthrie and Fink, 1991*). The strains used are isogenic either to BY4741 (*Mata ura3Δ0 his3Δ1 leu2Δ0 met15Δ0*), BY4742 (*Matα his3Δ1 leu2Δ0 lys2Δ0 ura3Δ0*), or FY251 (*Mata ura3-52 his3Δ200 leu2Δ1 trp1Δ63*) and are listed in *Supplementary file 1A*. Plasmids and primers used in this study are listed in *Supplementary files 1B and C*, respectively.

*ERG1* gene (including endogenous promoter and terminator regions) was cloned by PCR amplification from genomic DNA isolated from the strain BY4741. Primers were designed to anneal approximately 500 bp upstream and downstream of *ERG1* ORF and to introduce NotI and XhoI restriction sites for cloning in the polylinker of pRS316 plasmid originating the plasmid pPC943. Erg1-derivatives with specific lysine to arginine mutations were made by site-directed mutagenesis using pPC943 as template (*Supplementary file 1B*). To replace the endogenous copy of Erg1 with the lysine mutant versions, the C-terminal portion of Erg1 was amplified by PCR from pPC943 including the Erg1 3′UTR region. This fragment was digested with XbaI and NotI restriction sites (introduced by the primers) and cloned in a pRS303 plasmid originating pPC995. The resulting plasmid was linearized with EcoRI and integrated into the *ERG1* locus of the recipient strain. Transformants were selected based on HIS prototrophy and confirmed by DNA sequencing. In all cases, the recombinant plasmids were able to restore sterol synthesis, with the engineered Erg1 protein as the sole source of Erg1 activity. Wild type strain BY4741 was transformed by PCR-based homologous recombination to introduce the *GAL* promoter and HA-tag upstream of *ERG1* ORF generating strain yPC6475. PCR product was obtained from amplification of plasmid pYM-N24 (*Janke et al., 2004*) using primers 1242 and 1243. Strain yPC6475 was transformed with plasmids encoding Erg1-derivatives with specific lysine to arginine mutations as specified in *Supplementary files 1A, B*. The plasmid encoding for the misfolded protein Pca1$_{1–392}$-DHFR-HA (pPC860) was generated by cloning the DNA sequence coding for FLAG-Pca1$_{1–392}$ under the constitutive *PRC1* promoter into pRS316 and a sequence coding *Escherichia coli* DHFR was fused in frame, followed by a 3× HA tag-coding sequence and the *PRC1* terminator.

The plasmid encoding for the modular misfolded protein Vma12-Ndc10$_{902–956}$-HA (pPC926) was generated by cloning the sequence coding full-length Vma12 fused to a fragment of Ndc10 that is sufficient to promote its Doa10-dependent degradation, Ndc10$_{902–956}$ (*Furth et al., 2011*), and to a GGS–3× HA tag-coding sequence. The resulting fusion protein was expressed under the constitutive *PRC1* promoter from the plasmid pRS316.

## SILAC labeling, protein extraction, and proteomics

For SILAC experiments, the yeast strain BY4742, which is an auxotroph for lysine, was used to generate the strains *doa10Δ, ubc6Δ, ubc7Δ, hrd1Δ, usa1Δ*, and *der1Δ* by PCR-based homologous recombination. Cells were grown in 5 ml of synthetic complete media containing either 1 mM L-lysine (light) or 1 mM L-lysine [$^{13}C_6^{15}N_2$] (heavy) until stationary phase. Cultures were then diluted to OD600 ~0.005 in 50 ml fresh medium of the same composition. Cells were harvested at OD600 of 1.0–1.2. Then 20 OD of cells grown in light lysine media were mixed with 20 OD of cells grown in heavy lysine, harvested by centrifugation, and washed twice with cold water. Total cell extracts were prepared by glass-bead disruption in lysis buffer (6 M urea, 50 mM Tris–Cl pH 8.0, 0.5% SDS, 0.5% NP-40, 10 mM DDT). After removal of cell debris by centrifugation at 2000×*g*, proteins were precipitated by the addition of TCA to a final concentration of 20%. The protein pellet was washed with ice-cold acetone which was removed by drying the pellet at 50°C. Proteins were resuspended in a small amount of buffer (6 M urea, 50 mM Tris–Cl pH 7.5) and quantified by the BCA quantification method (Bio-Rad, Hercules, CA). Urea concentration was reduced to 4 M and the pH was adjusted to 8.8 with 25 mM Tris–HCl. Samples were digested with Lys-C in a 1:10 enzyme–protein ratio and desalted using an Oasis Plus HLB cartridge (Waters, Milford, MA). Digestions were fractionated using electrostatic repulsion-hydrophilic interaction chromatography (ERLIC). Each peptide fraction was separated by nanoLC in an EasyLC system (Proxeon) prior to mass spectrometric analysis on an LTQ-Orbitrap Velos Pro (Thermo Fisher Scientific, Waltham, MA) fitted with a nanospray source (Thermo Fisher Scientific). Data analysis was performed using the Proteome Discoverer software suite (v1.3.0.339; Thermo Fisher Scientific) and the Mascot search engine (v2.3; Matrix Science) was used for peptide identification. Data were searched against an in-house generated database containing all proteins in the *Saccharomyces* Genome Database plus the most common contaminants. The identified peptides were filtered using an FDR lower than 1%. Peptide areas were used to calculate the heavy–light ratios.

## Cycloheximide shut-off experiments and drug treatments

Cycloheximide shut-off experiments were performed in exponentially growing cells, as described (*Carvalho et al., 2010*). *cdc48-3* and *npl4-1* temperature sensitive strains were grown in synthetic complete medium at permissive temperature (25°C) to exponential phase and then shifted to 37°C for 2 hr before addition of cycloheximide. Sterol synthesis inhibitors were added to culture medium 2 hr (ZA and Ro-48) or 1 hr (Fluco) before addition of cycloheximide. DMSO at the same final concentration was added to control strains under the same experimental conditions. For each time point, samples corresponding to 1 OD of yeast cells were collected and protein extracts were prepared and analyzed as previously described (*Carvalho et al., 2010*).

## Mass spectrometric lipid analysis

Quantification of lipid species was performed essentially as previously described (*Ejsing et al., 2009*; *Klose et al., 2012*). In short, yeast cell pellets were resuspended in 1 ml 155 mM ammonium acetate and disrupted by vigorous shaking in the presence of 300 µl glass beads in 1.5 ml Eppendorf tubes. Cell lysates equivalent to 0.4 OD600 in 200 µl 155 mM ammonium acetate were spiked with a cocktail of internal lipid standards followed by two-step lipid extraction. Lipid extracts were analyzed using a LTQ Orbitrap XL mass spectrometer (Thermo Fisher Scientific) equipped with a robotic TriVersa NanoMate ion source (Advion Biosciences, Ithaca, NY). Lipid species were identified and quantified using MSFileReader (Thermo Fisher Scientific), ALEX software, and Orange software 2.6 (*Curk et al., 2005*). Lipid species were annotated using sum composition nomenclature as previously described (*Klose et al., 2012*).

## Cell culture and RNA interference

Tissue culture cells were grown in monolayer at 37°C in an atmosphere of 8–9% $CO_2$. HEK293 cells were cultured in DMEM high-glucose medium (Dulbecco's modified Eagle's medium containing 100 U/ml penicillin and 100 mg/ml streptomycin sulfate) supplemented with 10% (vol/vol) fetal calf serum (FCS). Cells were seeded in 6-well plates at $2 \times 10^5$ cells/well. The next day, they were transfected using Fugene HD reagent (Promega, Madison, WI) with a DNA–transfection reagent ratio of 1:3, according to the manufacturer's instructions. Myc-tagged TEB4- or TEB4(C9A)-encoding plasmids were previously described in *Hassink et al. (2005*). On the second day after transfection, cells were sterol deprived by an overnight statin treatment as described below.

RNAi was carried out as described previously with minor modifications (*Sever et al., 2003*). HEK293 cells were set up on day 1 at $2 \times 10^5$ cells/well in DMEM high-glucose medium containing 10% FCS. Cells were incubated with 20 µM of SMARTpool ON-TARGETplus MARCH6 siRNA (L-006925-00-0005) or 20 µM ON-TARGETplus Non-targeting Pool siRNA (D-001810-10-05) mixed with HiPerFect Transfection Reagent (Qiagen, Hilden, Germany) that was diluted in Opti-MEM I-reduced serum medium (Invitrogen, Carlsbad, CA) according to the manufacturer's instructions. After 24 hr incubation at 37°C, fresh DMEM high-glucose medium containing 10% FCS was added. On day 3, cells were incubated for 16 hr at 37°C in DMEM high-glucose medium containing 5% fetal bovine lipoprotein-deficient serum (LPDS), the HMGR inhibitor compactin (5 µM), and a low level of mevalonate (50 µM) that allows synthesis of essential non-sterol isoprenoids but not of cholesterol (*Hartman et al., 2010*). After 16 hr incubation, residual compactin was removed by washing the cells with PBS. Cells were treated as indicated in the figure legends. Test agents comprised 10 µg/ml cycloheximide, 20 µg/ml methyl-β-cyclodextrin (CD), or cholesterol/methyl-β-cyclodextrin complex (Chol/CD) (prepared as described previously in *Brown et al. [2002]*), and 10 µM MG-132 or an equivalent amount of DMSO. Cell lysates were prepared in 100 µl sample buffer (100 mM Tris–Cl pH 6.8, 3% SDS, 15% glycerol) after a wash step with PBS. Samples (usually 40 µg of proteins) were analyzed by 4–15% gradient SDS–PAGE and immunoblotted with the following antibodies: rabbit polyclonal anti-SM (1:5000) and mouse monoclonal anti-α-tubulin (1:1000). The relative intensities of bands were quantified using Quantity One software (Bio-Rad).

## Quantitative real-time PCR

RNA was harvested using the RNeasy RNA extraction kit (Qiagen) and reverse transcribed to yield complementary DNA (cDNA) with the SuperScript III First Strand cDNA Synthesis kit (Invitrogen). Levels of SM and MARCH6/Teb4 mRNA were determined relative to the housekeeping gene hypoxanthine phosphoribosyltransferase 1 (*HRPT1*) by quantitative real-time PCR using LightCycler 480 SYBR Green mix (Roche).

## Acknowledgements

We thank C Wang and E Wiertz for reagents, C Chiva and E Sabido for suggestions and help with the SILAC experiments, Z Berzina for help with lipid mass spectrometry, C Nogueira for help with the experiments in HEK cells, and Robert Oliete for generating Erg1 mutants. We thank A Palazzo, T Rapoport, and M Serra for critical reading of the manuscript.

## Additional information

### Funding

| Funder | Author |
| --- | --- |
| Center for Genomic Regulation (CRG) | Pedro Carvalho |
| Spanish MCCIN | Pedro Carvalho |
| International Early Career Scientist, Howard Hughes Medical Institute | Pedro Carvalho |
| Danish Council for Independent Research/Natural Sciences | Christer S Ejsing |
| Lundbeckfonden | Christer S Ejsing |
| La Caixa Graduate fellowship | Annamaria Ruggiano |

The funders had no role in study design, data collection and interpretation, or the decision to submit the work for publication.

### Author contributions

OF, Performed most of the experiments and analyzed the data; AR, Performed the experiments involving yeast growth assays; HKH-B, Performed lipid extractions and some lipid mass spectrometry; CSE, Performed most of the lipid mass spectrometry and analyzed the lipidomics data; PC, Conceived the study, analyzed and interpreted the data, and wrote the paper

## Additional files

### Supplementary files

• Supplementary file 1. (**A**) Yeast strains used in this study. (**B**) Plasmids used in this study. (**C**) Primers used in this study.

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
