## [Decision Letter]

Thank you for sending your work entitled “Sterol homeostasis requires regulated degradation of squalene monooxygenase by the ubiquitin ligase Doa10/Teb4” for consideration at *eLife*. Your article has been favorably evaluated by a Senior editor and 3 reviewers, one of whom is a member of our Board of Reviewing Editors.

The Reviewing editor and the other reviewers discussed their comments before we reached this decision, and the Reviewing editor has assembled the following comments to help you prepare a revised submission.

All three reviewers felt that the studies of yeast Erg1 are comprehensive and convincing. Moreover, they add a significant new element to our understanding of the regulation of sterol biosynthesis in this organism.

With regard to the animal cell experiments shown in Figure 5 and Figure 5—figure supplement 1, there is precedent for a role of cholesterol in accelerating degradation of squalene monooxygenase (SM). Nevertheless, we feel that the new aspect, namely the role of Teb4, is worthy of publication if it can be fully documented.

1) The experiments of Figure 5 show that elimination of Teb4 by SiRNA reduces, but does not abolish, the cholesterol-accelerated degradation of SM. Interpretation of this partial result would be aided if the authors quantitate the amount of Teb4 mRNA remaining after the SiRNA treatment. This is standard practice in SiRNA experiments. This one measurement would be the one piece of data that should be added prior to final acceptance.

2) In keeping with *eLife's* policy to minimize supplemental data, it will also be necessary for you to insert most of the supplementary figures into the body of the manuscript, editing the text accordingly.

3) You should also comply with the minor comments from Reviewer 3, listed below:

The model shown in Figure 6 should be revised to show that 24,25-dihydrolanosterol, not lanosterol, triggers degradation of mammalian HMG CoA reductase ([35] J. Biol. Chem. 284:26778 and [31] J. Biol. Chem. 283:1445). The model should also show that two ubiquitin ligases, gp78 and Trc8 (Jo et al. (2011) PNAS 108:20503), mediate degradation of mammalian HMG CoA reductase. Finally, the model should indicate that mevalonate-derived nonsterol isoprenoids contribute to mammalian HMG CoA reductase ERAD.

---

## [Author Response]

*1) The experiments of*
Figure 5
*show that elimination of Teb4 by SiRNA reduces, but does not abolish, the cholesterol-accelerated degradation of SM. Interpretation of this partial result would be aided if the authors quantitate the amount of Teb4 mRNA remaining after the SiRNA treatment. This is standard practice in SiRNA experiments. This one measurement would be the one piece of data that should be added prior to final acceptance*.

Treatment of Hek293 cells with siRNA directed to Teb4 lead to a 57% (± 0.044) reduction of TEB4 mRNA levels, as detected by qPCR. These data, originally in the legend of Figure 5, has now been incorporated within the main text.

*2) In keeping with* eLife's *policy to minimize supplemental data, it will also be necessary for you to insert most of the supplementary figures into the body of the manuscript, editing the text accordingly*.

We reduced the number of supplemental figures to four, as suggested. The data in these four figures is redundant with the results presented in the associated main figures and none of the conclusions of the paper are based on the supplemental figures. We believe these four figures should be presented as figure supplements.

*3) You should also comply with the minor comments from Reviewer 3, listed below*:

*The model shown in*
Figure 6
*should be revised to show that 24,25-dihydrolanosterol, not lanosterol, triggers degradation of mammalian HMG CoA reductase (*[35]
*J. Biol. Chem. 284:26778 and*
[31]
*J. Biol. Chem. 283:1445). The model should also show that two ubiquitin ligases, gp78 and Trc8 (Jo et al. (2011) PNAS 108:20503), mediate degradation of mammalian HMG CoA reductase. Finally, the model should indicate that mevalonate-derived nonsterol isoprenoids contribute to mammalian HMG CoA reductase ERAD*.

We incorporated the suggestions in Figure 6 and now acknowledge the role of Trc8 in HMGR degradation in the main text. We would like to thank the reviewer for helping us to provide a more complete and accurate model of the known roles of ERAD in sterol regulation.